# Community Correlations and Testing Independence Between Binary Graphs

## Abstract

Graph data has a unique structure that deviates from standard data assumptions, often necessitating modifications to existing methods or the development of new ones to ensure valid statistical analysis. In this paper, we explore the notion of correlation and dependence between two binary graphs. Given vertex communities, we propose community correlations to measure the edge association, which equals zero if and only if the two graphs are conditionally independent within a specific pair of communities. The set of community correlations naturally leads to the maximum community correlation, indicating conditional independence on all possible pairs of communities, and to the overall graph correlation, which equals zero if and only if the two binary graphs are unconditionally independent. We then compute the sample community correlations via graph encoder embedding, proving they converge to their respective population versions, and derive the asymptotic null distribution to enable a fast, valid, and consistent test for conditional or unconditional independence between two binary graphs. The theoretical results are validated through comprehensive simulations, and we provide two real-data examples: one using Enron email networks and another using mouse connectome graphs, to demonstrate the utility of the proposed correlation measures.

## 1 Introduction

Correlation or dependence quantifies the association between two random variables. Traditional Pearson's correlation (Pearson, 1895) measures linear association, while more recent methods, such as distance or kernel correlation (Székely et al., 2007; Gretton et al., 2005a), can detect any type of dependence. Since then, various new correlation and dependence measures have been introduced, each with their own advantages (Heller et al., 2013; Zhu et al., 2017; Shen et al., 2020; Pan et al., 2020; Chatterjee, 2021). Proper correlation or dependence measures can be very useful in a variety of downstream tasks, such as canonical correlation analysis (Hardoon et al., 2004; Tenenhaus & Tenenhaus, 2011; Shen et al., 2014), feature screening (Li et al., 2012; Zhong & Zhu, 2015; Zhang, 2024), time-series analysis (Zhou, 2012; Fokianos & Pitsillou, 2018; Shen et al., 2024a), conditional and partial testing (Fukumizu et al., 2007; Szekely & Rizzo, 2014; Wang et al., 2015), and two-sample testing (Panda et al., 2025), among others.

Typically, correlation and dependence measures and their properties are derived for standard random variables and the respective sample data. That is, given two random variables $(X, Y)$, each sample pair $(x_i, y_i)$ must be independently and identically distributed (i.i.d.). The i.i.d. assumption is crucial for the validity and consistency of sample data testing. However, many real-world data sets have inherent structures that violate this assumption. One notable example is time-series data, where testing independence via random permutations would be invalid, necessitating modifications to the testing process (Fokianos & Pitsillou, 2018; Shen et al., 2024a).

Another example is graph data, which has become increasingly popular in many scientific fields (Girvan & Newman, 2002; Choudhury & Mukherjee, 2009; Bullmore & Sporns, 2009; Leskovec & Krevl, 2014; Rossi & Ahmed, 2015; Hu et al., 2020). Given $n$ vertices and $s$ edges, a binary graph can be represented by an adjacency matrix $\mathbf{A} \in \{0, 1\}^{n \times n}$, where $\mathbf{A}(i, j) = 1$ indicates an edge between vertex $i$ and vertex $j$, and 0

otherwise. The i.i.d. assumption does not hold for graph data, e.g., $\mathbf{A}(i, j) = \mathbf{A}(j, i)$ for undirected graphs, and $\mathbf{A}(i, i) = 0$ for graphs without self-loops.

As the collection of graph data accelerates, the notion and availability of multi-graph data become increasingly common, ranging from multiplex network modeling to multi-graph embedding and analysis on multiple matched graphs (Domenico et al., 2013; Kivelä et al., 2014; Stella et al., 2015; 2017; Kinsley et al., 2020; Jones & Rubin-Delanchy, 2021; Shen et al., 2024c). In such settings, one may have multiple graphs $\mathbf{A}_1, \mathbf{A}_2, \ldots$, where the vertex set is matched across all graphs, while the edges may be independently generated or related across the graphs. A natural question is how to properly define a correlation measure between two graphs and what such a correlation means.

A common strategy for handling graph data is through graph embedding. A theoretically sound choice is spectral embedding, which provides a low-dimensional Euclidean representation (Rohe et al., 2011; Sussman et al., 2012) and can be proven to converge to the latent position of each vertex under the random graph model (Sussman et al., 2014; Athreya et al., 2018; Rubin-Delanchy et al., 2022). Since the latent positions can be assumed to be i.i.d., Pearson correlation, distance correlation, etc., can be directly applied to the spectral embedding to quantify the relationship between latent positions. This strategy has been used in two-sample testing between two graphs (Tang et al., 2017b;a) and in testing the independence between graphs and attributes (Lee et al., 2019). It has also been shown that the spectral radius can be used as a measure of graph correlation (Fujita et al., 2017).

While the spectral embedding approach is versatile, it has certain limitations: firstly, the resulting correlation or dependence measures the associations between the latent positions rather than directly between the two graphs; secondly, spectral embedding requires a selection of embedding dimensions, which is typically estimated on sample data and can be inaccurate; thirdly, spectral embedding can be relatively slow due to its use of singular value decomposition. Therefore, it is natural to wonder whether a proper correlation measure can be defined directly between two graphs, and how to compute the sample correlation efficiently. To the best of our knowledge, there has been little exploration of graph correlation, with the closest investigation being the use of correlated random graph models for seeded graph matching (Lyzinski et al., 2014; 2016).

In this paper, we aim to address this important gap by using community correlation to quantify the association between two graphs. We first review the stochastic block model, a classical community-based random graph model. This model helps motivate and understand the dependence structure between two graphs, showing that two graphs can be either unconditionally independent or, more often, conditionally independent via the vertex communities. We then propose a simple and intuitive correlation measure called community correlation, based on conditional probabilities via vertex communities, and prove that the community correlation equals zero if and only if two graphs are independent conditional on a specific pair of communities. This naturally leads to the maximum community correlation, which equals zero if and only if two graphs are independent conditional on any pair of communities, as well as to an overall graph correlation without using community information, which equals zero if and only if the two binary graphs are unconditionally independent. Therefore, the definition of community correlation provides a flexible and unified framework that can detect both conditional and unconditional dependence between two binary graphs.

We then compute the sample community correlations, sample maximum community correlation, and sample graph correlation via the graph encoder embedding algorithm (Shen et al., 2023b) as a proxy. We prove that each sample statistic converges to the corresponding population correlation and derive the asymptotic null distribution. As a result, the testing process is not only fast but also asymptotically valid and consistent for testing either conditional or unconditional independence between two binary graphs. The simulation sections consider various types of graph dependencies based on the stochastic block model to understand and illustrate the advantages of the proposed community correlation. We then use our proposed method to test graph independence for two real data sets: the Enron email networks and the mouse connectome graphs. All proofs are in the appendix.

## 2 Defining Graph Dependence

We first review the stochastic block model (SBM), followed by the definitions of binary graph variables and the definition of independence between two graph variables. Note that for presentation purposes, we always assume undirected graphs without self-loops in this paper. However, the methods and theory are also applicable to directed graphs with minimal changes.

### 2.1 The Stochastic Block Model

The stochastic block model is a widely-used random graph model known for its simplicity and ability to capture community structures (Holland et al., 1983; Snijders & Nowicki, 1997; Karrer & Newman, 2011). Under SBM, each vertex $i$ is assigned a class label $\mathbf{Y}(i) \in \{1, \ldots, K\}$, which may be fixed priori or assumed to follow a categorical distribution with prior probabilities $\{\pi_k \in (0, 1], \sum_{k=1}^{K} \pi_k = 1\}$.

Given the vertex labels, the SBM independently generates each edge between vertex $i$ and another vertex $j > i$ using a Bernoulli random variable:

$$\mathbf{A}(i, j) \sim \text{Bernoulli}\{B(\mathbf{Y}(i), \mathbf{Y}(j))\}.$$

Here, $B = [B(k, l)] \in [0, 1]^{K \times K}$ represents the block probability matrix, which serves as the parameters of the model. For undirected graph without self-loop, $\mathbf{A}(j, i) = \mathbf{A}(i, j)$ and $\mathbf{A}(i, i) = 0$.

Many real-world graphs are heterogeneous, with different vertices having varying degrees, and the graph can be very sparse. To accommodate this, the degree-corrected stochastic block model (DC-SBM) was introduced as an extension of SBM (Zhao et al., 2012): in addition to the existing parameters of SBM, DC-SBM adds an additional degree vector $[\theta(1), \theta(2), \ldots, \theta(n)] \in \mathbb{R}^n$, where each element $\theta(i)$ controls the degree of vertex $i$. Then the edge between vertex $i$ and another vertex $j \neq i$ is independently generated by:

$$\mathbf{A}(i, j) \sim \text{Bernoulli}\{\theta(i)\theta(j)B(\mathbf{Y}(i), \mathbf{Y}(j))\}.$$

Typically, degrees may be assumed to be fixed a priori or independently and identically distributed per vertex.

Overall, one may define a binary graph variable $A$ between two vertices as a mixture of Bernoulli distribution to capture SBM:

$$A \sim \sum_{k,l=1,\ldots,K} I(Y = k)I(Y' = l)\text{Bernoulli}\{\theta\theta'B(k, l)\}$$

where $Y$ is the categorical variable for the first vertex, $I(\cdot)$ is the 0-1 indicator function, $I(Y = k)$ equals 1 with probability $\pi_k$, and $Y'$ is an independent copy of $Y$ for the second vertex. Then each $\mathbf{A}_{ij}$ for $i < j$ is independently and identically distributed as $F_A$.

### 2.2 Conditional and Unconditional Independence between Graphs

Given two sample graphs $\mathbf{A}_1 \in \{0, 1\}^{n \times n}$ and $\mathbf{A}_2 \in \{0, 1\}^{n \times n}$ from the stochastic block model with the same block matrix $B$, suppose the vertices are matched, and each pair of $(\mathbf{A}_1(i, j), \mathbf{A}_2(i, j))$ is independently and identically distributed as a pair of binary graph variables $(A_1, A_2)$ for all $i < j$. When the vertices share the same communities, and the edge generations are independent, we have

$$A_1 \sim \sum_{k,l=1,\ldots,K} I(Y = k)I(Y' = l)\text{Bernoulli}\{B(k, l)\},$$

$$A_2 \sim \sum_{k,l=1,\ldots,K} I(Y = k)I(Y' = l)\text{Bernoulli}\{B(k, l)\},$$

where the two Bernoulli variables are independent. Despite the edges being independent conditioned on $(Y, Y')$, $A_1$ and $A_2$ are actually dependent via $(Y, Y')$.

This example shows that when two graphs have the same or dependent vertex communities, unconditional independence becomes trivially false, and conditional independence is what actually matters in assessing whether the edges are dependent or not. Graphs with shared or highly related communities are very common in practice. For example, in brain graphs, each community represents a brain region with common functionality across people. This motivates the following definition of conditional independence:

**Definition 1.** *We say two graphs are conditionally independent on a specific pair $(Y, Y') = (k, l)$, if and only if the joint distribution of the graph variables satisfies:*

$$F_{A_1, A_2|(Y,Y')=(k,l)}(a_1, a_2) = F_{A_1|(Y,Y')=(k,l)}(a_1) F_{A_2|(Y,Y')=(k,l)}(a_2). \tag{1}$$

*And we say two graphs are conditionally independent on all possible values of $(Y, Y')$, if and only if Equation 1 holds for all possible $(k, l) = \{1, \ldots, K\}^2$.*

For two SBM graphs to be unconditionally independent, excluding the trivial case where all entries in the block matrix are the same, a necessary requirement is that the two graphs have independent vertex communities, e.g.,

$$A_1 \sim \sum_{k,l=1,\ldots,K} I(Y = k) I(Y' = l) \text{Bernoulli}\{B(k,l)\},$$

$$A_2 \sim \sum_{k,l=1,\ldots,K} I(Y'' = k) I(Y''' = l) \text{Bernoulli}\{B(k,l)\},$$

where all of $Y, Y', Y'', Y'''$ must be independent. Therefore, unconditional independence can occur between two graphs but is relatively rare, as it requires that two vertices, despite being matched in context, have independent community structures across graphs. A formal definition of unconditional independence is as follows:

**Definition 2.** *We say two graphs are unconditionally independent if and only if the joint distribution of the graph variables satisfies:*

$$F_{A_1, A_2}(a_1, a_2) = F_{A_1}(a_1) F_{A_2}(a_2).$$

Note that both definitions are grounded in the standard notion of probabilistic independence, but are formulated in terms of a pair of graph-valued random variables.

### 2.3 Motivation

To better motivate the importance of considering both unconditional and conditional dependence, Figure 1 presents a simple analogy using standard random variables and Pearson correlation.

In the left panel, $A_1$ and $A_2$ exhibit high unconditional correlation, but this association is largely explained by their shared dependence on latent variables. After conditioning on both $Y$ and $Y'$, the correlation drops to around zero — indicating that the apparent relationship between $A_1$ and $A_2$ is spurious and induced by a common latent factor, which is particularly relevant in causal inference.

In contrast, the right panel illustrates a situation where the conditional correlation is stronger than the unconditional one. Here, conditional dependence between $A_1$ and $A_2$ is masked when averaging across groups by $Y$ and $Y'$. This highlights the need to examine conditional dependence even if the unconditional dependence appears weak.

## 3 Population Correlations

We define the community correlation, the maximum community correlation, and the graph correlation based on population probabilities given a pair of binary graph variables.

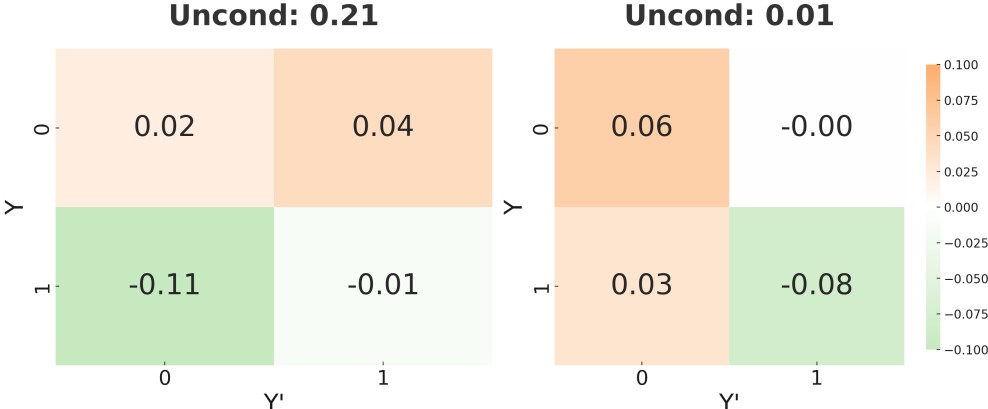

Figure 1: This figure compares the unconditional Pearson correlation $PCor(A_1, A_2)$ and conditional correlation $PCor(A_1, A_2|Y, Y')$ based on standard random variable setting $A_1, A_2$, and binary $Y$ and $Y'$. The unconditional correlation is displayed in the title of each panel, while the conditional correlations are shown within each $(Y, Y')$.

**Community Correlation**

Given a pair of binary graph variables $(A_1, A_2)$, and a pair of categorical community variable $(Y, Y') \in [1, 2, \ldots, K]^2$. We define the population community covariance as:

$$\Sigma_{12}(Y, Y') = \text{Prob}(A_1 = 1, A_2 = 1|Y, Y') - \text{Prob}(A_1 = 1|Y, Y')\text{Prob}(A_2 = 1|Y, Y'),$$
$$\Sigma_1(Y, Y') = \text{Prob}(A_1 = 1|Y, Y') - \text{Prob}(A_1 = 1|Y, Y')^2,$$
$$\Sigma_2(Y, Y') = \text{Prob}(A_2 = 1|Y, Y') - \text{Prob}(A_2 = 1|Y, Y')^2.$$

The population community correlation follows by

$$\rho(k, l) = \frac{\Sigma_{12}(k, l)}{\sqrt{\Sigma_1(k, l)\Sigma_2(k, l)}} \in [-1, 1].$$

The correlation is supported in $[-1, 1]$ by applying the Cauchy-Schwarz inequality. If the denominator equals 0, then $\rho(k, l)$ is also set to 0. Therefore, the community correlation $\rho(k, l)$ measures the strength of relationship within a specific pair of community $(k, l)$.

**Maximum Community Correlation**

There are $K \times K$ such community correlations. In the case of two undirected graphs, we have $\rho(k, l) = \rho(l, k)$, so effectively there are $K(K + 1)/2$ different community correlations, based on which we can define the maximum community correlation:

$$\rho^m = \max_{k, l=1, \ldots, K} |\rho(k, l)|,$$

which measures the maximum strength of the relationship conditioned on all possible values of $(Y, Y')$. While it is possible to use other variants, such as mean correlation or multiple testing, the maximum statistic has been shown to be more advantageous in finite-sample testing power (Shen & Dong, 2024).

**Graph Correlation**

Similarly, we define the graph covariance as:

$$\Sigma_{12} = \text{Prob}(A_1 = 1, A_2 = 1) - \text{Prob}(A_1 = 1)\text{Prob}(A_2 = 1),$$
$$\Sigma_1 = \text{Prob}(A_1 = 1) - \text{Prob}(A_1 = 1)^2,$$
$$\Sigma_2 = \text{Prob}(A_2 = 1) - \text{Prob}(A_2 = 1)^2.$$

and the graph correlation as:

$$\rho = \frac{\Sigma_{12}}{\sqrt{\Sigma_1 \Sigma_2}} \in [-1, 1].$$

which measures the strength of relationship between $A_1$ and $A_2$ without any conditioning.

**Population Properties**

The population correlations defined so far not only measure the strength of the relationship, but are also proper dependence measures for testing different types of independence between two graph variables:

**Theorem 1.** *Given a pair of binary graph variables $(A_1, A_2)$, the following holds:*

- *$\rho(k, l) = 0$ if and only if $A_1$ and $A_2$ are conditionally independent on a specific pair $(Y, Y') = (k, l)$.*

- *$\rho^m = 0$ if and only if $A_1$ and $A_2$ are conditionally independent on all possible values of $(Y, Y')$.*

- *$\rho = 0$ if and only if $A_1$ and $A_2$ are unconditionally independent.*

## 4 Sample Method and Consistency

Given the sample graphs, here we present the sample algorithm for computing community correlations and conducting hypothesis testing, followed by discussions on several extensions and practical considerations. We then address the sample convergence properties and the validity and consistency of the sample tests.

### 4.1 Sample Community Correlations and Independence Testing

- **Input**: Two sample graphs $\mathbf{A}_1, \mathbf{A}_2 \in \mathbb{R}^{n \times n}$, a label vector $\mathbf{Y} \in \{1, \dots, K\}^n$, and a positive threshold $\epsilon > 0$.

- **Step 1**: Calculate the number of observations per class, denoted as:

$$n_k = \sum_{i=1}^{n} 1(\mathbf{Y}(i) = k)$$

for $k = 1, \dots, K$. Then construct the normalized one-hot matrix $\mathbf{W} \in [0, 1]^{n \times K}$ as follow: for each observation $i = 1, \dots, n$, set

$$\mathbf{W}(i, k) = 1/n_k$$

if and only if $\mathbf{Y}_i = k$, and 0 otherwise.

- **Step 2 (Graph encoder embedding)**: Compute the graph encoder embedding for $\mathbf{A}_1, \mathbf{A}_2$, and $\mathbf{A}_1 \odot \mathbf{A}_2$ as follows:

$$\mathbf{Z}_{12} = (\mathbf{A}_1 \odot \mathbf{A}_2)\mathbf{W},$$
$$\mathbf{Z}_1 = \mathbf{A}_1 \mathbf{W},$$
$$\mathbf{Z}_2 = \mathbf{A}_2 \mathbf{W},$$

where $\odot$ denotes the entry-wise product.

- **Step 3 (Community Correlations)**: For each $k, l = 1, \ldots, K$, compute the sample covariance and variance terms for each community pair $(k, l)$ as

$$\hat{\Sigma}_{12}(k,l) = \sum_{\substack{i=1,\ldots,n}}^{\mathbf{Y}(i)=k} \frac{\mathbf{Z}_{12}(i,l)}{n_k} - \sum_{\substack{i=1,\ldots,n}}^{\mathbf{Y}(i)=k} \frac{\mathbf{Z}_1(i,l)}{n_k} \sum_{\substack{i=1,\ldots,n}}^{\mathbf{Y}(i)=k} \frac{\mathbf{Z}_2(i,l)}{n_k},$$

$$\hat{\Sigma}_1(k,l) = \sum_{\substack{i=1,\ldots,n}}^{\mathbf{Y}(i)=k} \frac{\mathbf{Z}_1(i,l)}{n_k} - \sum_{\substack{i=1,\ldots,n}}^{\mathbf{Y}(i)=k} \frac{\mathbf{Z}_1(i,l)}{n_k} \sum_{\substack{i=1,\ldots,n}}^{\mathbf{Y}(i)=k} \frac{\mathbf{Z}_1(i,l)}{n_k},$$

$$\hat{\Sigma}_2(k,l) = \sum_{\substack{i=1,\ldots,n}}^{\mathbf{Y}(i)=k} \frac{\mathbf{Z}_2(i,l)}{n_k} - \sum_{\substack{i=1,\ldots,n}}^{\mathbf{Y}(i)=k} \frac{\mathbf{Z}_2(i,l)}{n_k} \sum_{\substack{i=1,\ldots,n}}^{\mathbf{Y}(i)=k} \frac{\mathbf{Z}_2(i,l)}{n_k},$$

  followed by computing the community correlation

$$\hat{\rho}(k,l) = \frac{\hat{\Sigma}_{12}(k,l)}{\sqrt{\hat{\Sigma}_1(k,l)\hat{\Sigma}_2(k,l)}},$$

  which is set to 0 if the denominator equals 0.

- **Step 4 (Maximum Community Correlation)**: Compute the sample maximum correlation, but exclude all off-diagonal community correlations whose corresponding variance is too small:

$$\hat{\rho}^m = \max_{\substack{k,l=1,\ldots,K \\ k \neq l}} \{ \frac{\sqrt{n_k n_l}}{n} |\hat{\rho}(k,l)| \cdot 1(\hat{\Sigma}_1(k,l) \geq \epsilon) 1(\hat{\Sigma}_2(k,l) \geq \epsilon) \}\},$$

$$q = \sum_{k,l=1,\ldots,K} 1(\hat{\Sigma}_1(k,l) < \epsilon) 1(\hat{\Sigma}_2(k,l) < \epsilon),$$

  where $q$ counts how many community correlations are excluded.

- **Step 5 (Testing Conditional Independence)**: Compute the p-value by:

$$\text{pval} = 1 - (2\text{Prob}(n|\hat{\rho}^m| \geq \text{Normal}(0,2)) - 1)^{K(K+1)/2-q/2}. \tag{2}$$

- **Output**: $\{\hat{\rho}(k,l)\}, \hat{\rho}^m, \text{pval}$.

The overall p-value determines whether two graphs are conditionally independent on all possible values of $(Y, Y')$. When the p-value is smaller than a pre-set type 1 error level $\alpha$, the conditional independence hypothesis is rejected. Note that this is a two-sided test, and in the experiment, we default $\epsilon = 0.01$. Also note that if either graph is directed, then the exponent in Equation 2 should be $K^2 - q$ instead.

In the following, we discuss several aspects of the proposed methods, including parameter selection, method choice, and computational complexity.

### 4.1.1 Thresholded Sample Maximum

In Step 4, we employed a small $\epsilon = 0.01$ to exclude correlations whose corresponding variance is too small. This is necessary mainly for small $n_k$ and sparse graphs, because when the number of edges is very few within the given $(k, l)$ pair, the sample variance can be very small, which may cause a large correlation just by chance. Such spurious large correlations can break the null distribution approximation and cause the testing power to be inflated. This is similar in concept to the smoothed maximum in (Vogelstein et al., 2019).

### 4.1.2 Within-Block Permutation as Alternative Test

In the sample method, we carry out the hypothesis testing by essentially assuming each $\frac{\sqrt{n_k n_l}}{n}\hat{\rho}(k,l)$ can be approximated by $\text{Normal}(0,2)$ in null distribution, which is a conservative estimate, followed by maximum order statistic.

Alternatively, one could use the standard permutation test (Good, 2005), which is commonly used in other methods for testing independence. However, to ensure valid testing, the permutations need to be carried out within each community. For example, suppose there are 10 vertices where the first half is from community 1 and the second half is from community 2. The within-block permutation only permutes indices within each community, i.e., vertices $1, 2, 3, 4, 5$ can only be permuted within community 1. However, the computational complexity of the permutation test can be an issue for large graphs, as it requires 100 or more replicates to repeatedly compute the test statistic.

For small-sized or very sparse graphs, the permutation test can provide more accurate testing results. For moderate to large graphs where $n_k \geq 50$ for each community, the proposed test suffices for validity and consistency purposes (see Theorem 3) and is much faster than the permutation test.

### 4.1.3 Testing on Individual Pair of Communities

The above sample algorithm produces a p-value to test conditional independence on any pair of communities. To test on a specific pair $(k, l)$, one may take the individual community correlation and compute:

$$\text{pval}(k,l) = 2 - 2\text{Prob}(\sqrt{n_k n_l}|\hat{\rho}(k,l)| > \text{Normal}(0,2)).$$

Note that instead of using Step 4 and 5 to test conditional independence on any pair of communities, one could also use the $K \times K$ individual p-values followed by multiple testing (e.g., Bonferroni correction) to achieve valid testing. However, multiple testing can end up being too conservative for large $K$.

### 4.1.4 Testing Unconditional Independence via Sample Graph Correlation

The given sample method computes the maximum community correlation and tests conditional independence. By simply setting $\mathbf{Y} = 1$ for all vertices and carrying out step 1 to 3, the method outputs the sample graph correlation $\hat{\rho}$. The resulting p-value can be computed by

$$\text{pval}(k,l) = 2 - 2\text{Prob}(n|\hat{\rho}| > \text{Normal}(0,2)).$$

We reject the unconditional independence hypothesis when the p-value is smaller than $\alpha$. Therefore, the computation steps are essentially the same, except $K = 1$ in this case.

### 4.1.5 On Graph Encoder Embedding

We shall note that the usage of graph encoder embedding in this context is optional. The sample correlations can, in fact, be directly computed between two graph adjacency matrices, where steps 1-3 can be modified so that the sample covariance and variances are computed via double summations of the graph adjacency matrices (see proof of Theorem 2 for details). However, the usage of encoder embedding significantly simplifies the notation and presentation, and makes the sample definition more in line with traditional correlation measures.

### 4.1.6 On Label Vector

In cases where no label is available but one would like to test conditional independence, there exist abundant community detection methods that can be applied to estimate vertex labels, such as the Leiden algorithm, spectral clustering, or likelihood-based community detection (Traag et al., 2019; Zhao et al., 2012; Abbe, 2018; Lei et al., 2020; Mu et al., 2022; Shen et al., 2023a). Once an estimated label vector is obtained, conditional testing can then be performed.

### 4.1.7 Computational Complexity

Step 1 to 3 have a complexity of $O(n^2)$ for dense graphs, and $O(nK + s)$ for sparse graphs or edgelist, where $s$ is the number of edges in both graphs. Step 4 iterates through all community correlations, which takes $O(K^2)$. Step 5 computes the p-value by comparing the maximum dependence measure to its approximate null distribution, which is also $O(K^2)$. Since sparse graphs are common in social networks, the average complexity is $O(nK + s + K^2)$. Given that $K$ is typically a small constant, the number of edges $s$ becomes the dominant term, simplifying the average complexity to $O(s)$ for sparse graphs. For dense graphs or when $K$ is on the order of $O(n)$, the worst-case complexity reaches $O(n^2)$.

### 4.2 Consistency of Sample Estimator

**Theorem 2.** *As the number of vertices $n \to \infty$, assume all $n_k$ also converge to infinity, and the threshold $\epsilon \to 0$. Then, all the sample covariance and variance estimators are consistent estimators of their population counterparts:*

$$\hat{\Sigma}_{12}(k, l) \overset{n \to \infty}{\to} \Sigma_{12}(k, l),$$
$$\hat{\Sigma}_{1}(k, l) \overset{n \to \infty}{\to} \Sigma_{1}(k, l),$$
$$\hat{\Sigma}_{2}(k, l) \overset{n \to \infty}{\to} \Sigma_{2}(k, l).$$

*As a result, the sample community correlation converges to the population correlation:*

$$\hat{\rho}(k, l) \overset{n \to \infty}{\to} \rho(k, l),$$

*the sample maximum correlation is a consistent estimator of the population maximum:*

$$\hat{\rho}^m \overset{n \to \infty}{\to} \rho^m,$$

*and the sample graph correlation also converges to the population graph correlation:*

$$\hat{\rho} \overset{n \to \infty}{\to} \rho.$$

From Theorem 1 and Theorem 2, each sample correlation is consistent for the corresponding hypothesis test.

**Corollary 1.** *Under the same assumptions as Theorem 2, we have:*

- *$\hat{\rho}(k, l) \to 0$ if and only if $A_1$ and $A_2$ are conditionally independent on a specific pair $(Y, Y') = (k, l)$.*

- *$\hat{\rho}^m \to 0$ if and only if $A_1$ and $A_2$ are conditionally independent on all possible values of $(Y, Y')$.*

- *$\hat{\rho} \to 0$ if and only if $A_1$ and $A_2$ are unconditionally independent.*

Note that the condition in Theorem 2 simply means that as the number of vertices increases, the number of vertices per community shall also increase together with $n$; and as the number of vertices becomes sufficiently large, the threshold $\epsilon$ is effectively set to 0 and not required any more to prune any sample community correlations.

### 4.3 Asymptotic Null Distribution and Validity

Finally, we show the proposed sample test is asymptotically valid. To achieve this, it suffices to show the null distribution of the sample community correlation, then the distribution of the maximum community correlation follows by order statistics, and the distribution of the sample graph correlation follows as a special case where $k = l = K = 1$.

**Theorem 3.** *Let $n$ increase to infinity, and assume the underlying binary graph variables $(A_1, A_2)$ are conditionally independent on $(Y, Y')$. When $k \neq l$, the sample community correlation satisfies*

$$\sqrt{n_k n_l} \hat{\rho}(k, l) \overset{dist}{\to} Normal(0, 1)$$

with a difference of at most $O(\frac{1}{n})$, i.e., denote $U = \sqrt{n_k n_l}\hat{\rho}(k,l)$ and $V$ be the standard normal variable, their distribution satisfies:

$$|f_U(x) - f_V(x)| = O(\frac{1}{n}).$$

When $k = l$, the sample community correlation satisfies the same, except it converges to $Normal(0, \sqrt{2})$, i.e.,

$$n_k\hat{\rho}(k,k) \stackrel{dist}{\rightarrow} Normal(0, \sqrt{2}).$$

The normal distribution is the asymptotic distribution for large $n$, which can be slightly aggressive for small $n$ (see Figure 2). Therefore, the sample method uses $Normal(0, 2)$ as a conservative null distribution estimate for testing, which ensures validity and consistency of the resulting independence test (see Figure 4 and 5), albeit slightly conservative for $n$ large. Interestingly, the null distribution is similar to the asymptotic null distribution of distance correlation in high dimensions (Székely & Rizzo, 2013; Shen et al., 2022).

## 5 Simulations

In the simulations, we first verify the null distribution of the community correlation and its computational scalability. We then validate the accuracy and consistency of our method for testing both conditional and unconditional independence across a range of SBM models with varying dependence structures. Additionally, we visualize the resulting community correlations and compute precision and recall in several representative scenarios. For testing power evaluation, we mainly compare with the spectral embedding (each graph is separately projected into $d = 20$) followed by distance correlation and partial distance correlation (Székely et al., 2007; Székely & Rizzo, 2014).

### 5.1 Evaluation of Null Distribution and Running Time

Figure 2 shows the empirical null distribution of the community correlations, using the same stochastic block model but with increasing sample size. The SBM graphs used are the conditionally independent simulation and the unconditionally independent simulation described in Section 5.2, with $n = 100, 500, 1000$, and $K = 10$. We run 1000 replicates to generate independent graphs and compute the community correlations as the empirical null distribution. It is clear that as the number of vertices and edges increases, the null distributions in both independent settings are very close to $Normal(0, 1)$ for $\sqrt{n_k n_l}\hat{\rho}(k,l)$ where $k \neq l$, and very close to $Normal(0, \sqrt{2})$ for $n_k\hat{\rho}(k,k)$, validating the asymptotic null distribution of the sample statistics. We also observe that the asymptotic null distributions can be slightly aggressive for small $n$, meaning using them for p-value computation may yield invalid tests that inflate the testing power for small $n$; on the other hand, $Normal(0, 2)$ is a conservative estimate that is generally valid for small $n$. Note that the empirical distribution for community correlation is similar regardless of conditional or unconditional independence. Additionally, when considering graph correlation under the unconditional independence scenario, the empirical null distribution matches that of $n_k\hat{\rho}(k,k)$.

Figure 3 shows the running time as the number of edges increases, using the first three SBM simulations in Section 5.2, and reports the average running time in log scale. Clearly, computing the community correlation (including all $K^2$ of them) is much faster than computing distance correlation or partial distance correlation on spectral embedding.

### 5.2 Evaluation of Testing Power under SBM

Here we consider the following six stochastic block model simulations, where $K = 10$, $\mathbf{Y} = [1, 2, \ldots, K]$ equally likely. The block matrix for the first graph is $B(k, l) = 0.1$ and $B(k, k) = 0.2$, with $\mathbf{A}_1(i, j) \sim$ Bernoulli$\{B(\mathbf{Y}(i), \mathbf{Y}(j))\}$ for $i < j$ throughout all models.

We then simulate the second graph $\mathbf{A}_2$ in six different ways.

## Cumulative Distribution Function

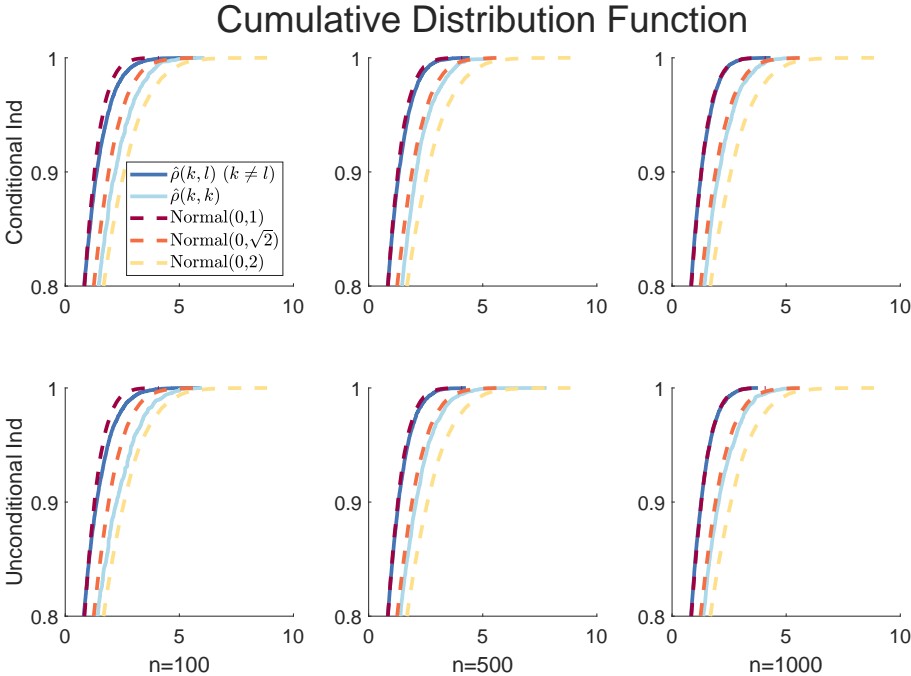

Figure 2: This figure shows the empirical null distribution of the community correlations, comparing it to three different normal distributions. The top row considers the conditionally independent simulation, while the bottom row considers the unconditionally independent simulation.

## Running Time

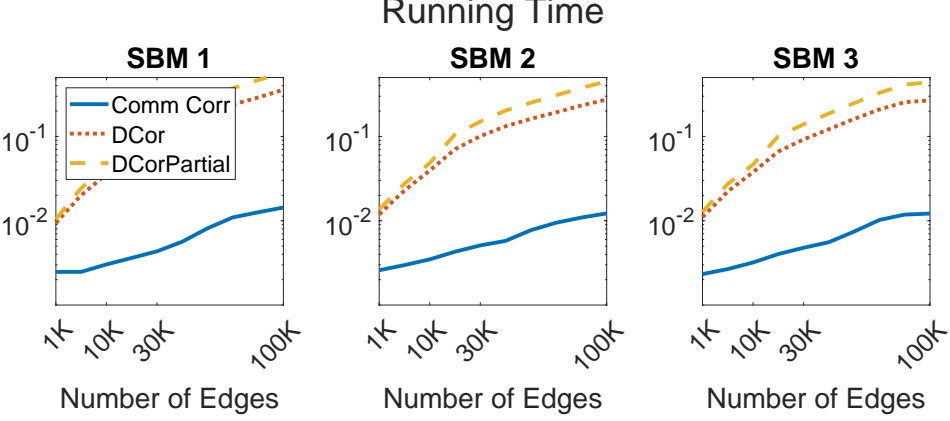

Figure 3: This figure shows the average running time of community correlation using 1000 replicates.

- Conditional independence: let

$$\mathbf{A}_2(i,j) \sim \text{Bernoulli}\{B(\mathbf{Y}(i), \mathbf{Y}(j))\}.$$

- Unconditional independence: let $\mathbf{Y}_2 = [1, 2, \ldots, K]$ be equally likely and independently generated from $\mathbf{Y}$, and

$$\mathbf{A}_2(i,j) \sim \text{Bernoulli}\{B(\mathbf{Y}_2(i), \mathbf{Y}_2(j))\}.$$

- All communities dependence: let

$$\mathbf{A}_2(i,j) \sim \text{Bernoulli}\{B(\mathbf{Y}(i), \mathbf{Y}(j)) + 0.1\mathbf{A}_1(i,j)\}.$$

- Community $(1, \cdot)$ dependence: let

$$\mathbf{A}_2(i,j) \sim \text{Bernoulli}\{B(\mathbf{Y}(i), \mathbf{Y}(j)) \\ + 0.1\mathbf{A}_1(i,j)1(Y(i) = 1))\}.$$

- Community $(1, 2)$ dependence: let

$$\mathbf{A}_2(i,j) \sim \text{Bernoulli}\{B(\mathbf{Y}(i), \mathbf{Y}(j)) \\ + 0.1\mathbf{A}_1(i,j)1(Y(i) = 1)1(Y(j) = 2))\}.$$

- Community $(1, 2)$ and $(5, 10)$ dependence: let

$$\mathbf{A}_2(i,j) \sim \text{Bernoulli}\{B(\mathbf{Y}(i), \mathbf{Y}(j)) \\ + 0.1\mathbf{A}_1(i,j)1(\mathbf{Y}(i) = 1)1(\mathbf{Y}(j) = 2) \\ + 0.1\mathbf{A}_1(i,j)1(\mathbf{Y}(i) = 5)1(\mathbf{Y}(j) = 10)\}.$$

All graphs are undirected and no self-loop. We compare the average testing power using maximum community correlation, graph correlation, distance correlation, and its partial variant on spectral embedding at a type 1 error level of 0.05. For all six simulations, we let $n$ increase, and plot the average testing power using 1000 replicates.

The first panel in Figure 4 shows that the maximum community correlation is the only correlation that is valid for testing conditional independence, as its testing power remains below 0.05. Moreover, the maximum community correlation is consistent for all types of graph dependence from panels 3 to 6, where the testing power converges to 1. While all other methods also have their testing power increasing to 1, their testing power is inflated due to being invalid for conditional independence.

For unconditional independence testing, however, all methods are valid and have a testing power no larger than 0.05, as shown in the second panel of Figure 4. Moreover, the proposed graph correlation has better testing power over spectral embedding in all cases, demonstrating the advantage of a proper correlation directly defined between two graphs.

## 5.3 Evaluation of Testing Power under DC-SBM

In this subsection, we consider the same evaluation as SBM, except we use DC-SBM models to make the graph more sparse and heterogeneous. Everything else remains the same, with $\theta_1$ and $\theta_2$ being two vectors where each entry is independently generated from $\text{Beta}(1, 2)$. The block matrix for the first graph is $B(k, l) = 0.1$ and $B(k, k) = 0.5$, with $\mathbf{A}_1(i,j) \sim \text{Bernoulli}\{\theta_1(i)\theta_1(j)B(\mathbf{Y}(i), \mathbf{Y}(j))\}$ for $i < j$ throughout all models. We then simulate the second graph $\mathbf{A}_2$ as follows.

- Conditional independence: let

$$\mathbf{A}_2(i,j) \sim \text{Bernoulli}\{\theta_2(i)\theta_2(j)B(\mathbf{Y}(i), \mathbf{Y}(j))\}.$$

- Unconditional independence: let $\mathbf{Y}_2 = [1, 2, \ldots, K]$ be equally likely and independently generated from $\mathbf{Y}$, and

$$\mathbf{A}_2(i,j) \sim \text{Bernoulli}\{\theta_2(i)\theta_2(j)B(\mathbf{Y}_2(i), \mathbf{Y}_2(j))\}.$$

- All Communities dependence: let

$$\mathbf{A}_2(i,j) \sim \text{Bernoulli}\{\theta_2(i)\theta_2(j)(B_1(Y(i), Y(j)) \\ + 0.3\mathbf{A}_1(i,j))\}.$$

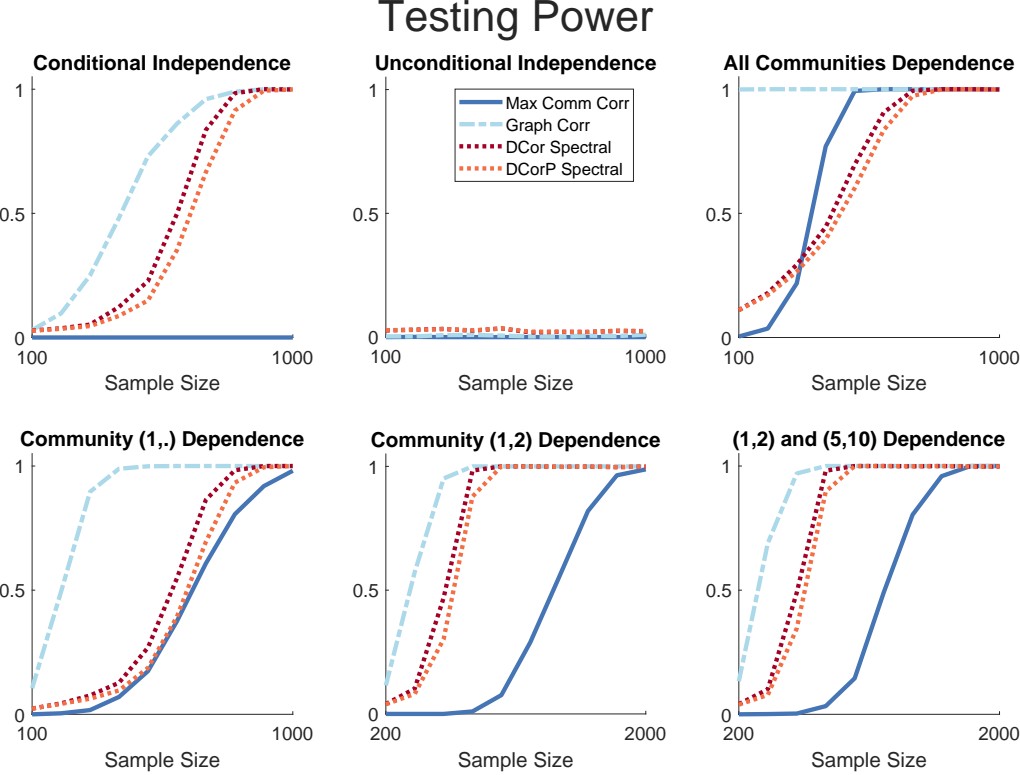

Figure 4: This figure evaluates the average testing power using 1000 replicates on six SBM simulations.

- Degree dependence: let

$$\mathbf{A}_2(i,j)\mathbf{A}_2(i,j) \sim \text{Bernoulli}\{\theta_1(i)\theta_1(j)B(\mathbf{Y}(i),\mathbf{Y}(j))\}.$$

Note that the first three simulations are similar to Section 5.2, and the last case is a degree dependence scenario unique to DC-SBM. The testing powers are reported in Figure 5, which has the same interpretation as Figure 4: the maximum correlation is valid and consistent for testing against conditional independence, while all other methods are valid and consistent for unconditional independence. Note that some dependency simulations in SBM are not repeated here, as the DC-SBM simulations performed very similarly, except the power convergence is slower here due to DC-SBM being sparser than SBM.

### 5.4 Visualization of Community Correlation

To show that the community correlation indeed quantifies the strength of the relationship, Figure 6 visualizes the community correlations for several simulations at $n = 1000$. With $K = 10$, there are $10 \times 10$ community correlations. The first panel shows that all community correlations are almost 0 (to two decimal places) in the case of conditional independence. The second panel shows that all correlations are almost 0.1, which matches the dependence model:

$$\mathbf{A}_2(i,j) \sim \text{Bernoulli}\{B(\mathbf{Y}(i),\mathbf{Y}(j)) + 0.1\mathbf{A}_1(i,j)\}$$

for which the strength of relationship is indeed 0.1, related via the Bernoulli probability. Moreover, the community correlations are also able to show when and how the two graphs are related. For example, only $\hat{\rho}(1,2)$ and $\hat{\rho}(2,1)$ are significant in the fourth panel. In the last panel, the degree dependence structure means there is higher dependence on the diagonal than off-diagonal correlations, which is also properly visualized.

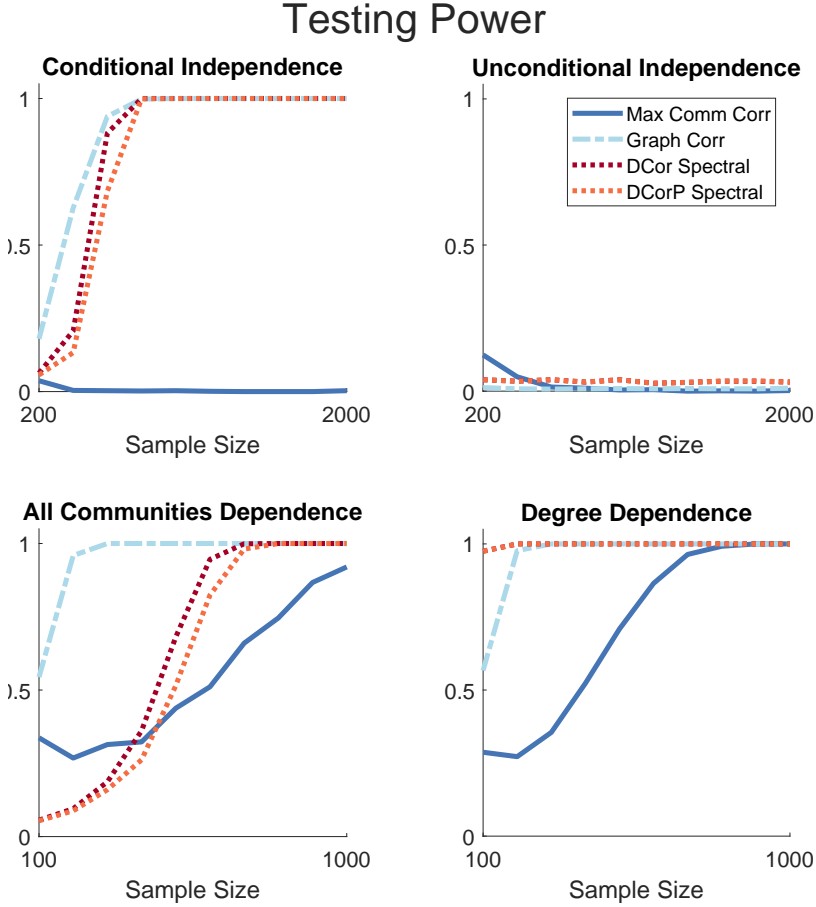

Figure 5: This figure evaluates the average testing power using 1000 replicates on four DC-SBM simulations.

### 5.5 Precision and Recall

To further demonstrate the performance of our method, we present the precision and recall results for three simulations from Section 5.2: community $(1, \cdot)$ dependence, community $(1, 2)$ dependence, and community $(1, 2)$ and $(5, 10)$ dependence. In each simulation, a few community pairs (e.g., 10, 1, or 2 pairs, respectively) are dependent, while all other pairs are independent, which enables a meaningful evaluation of precision and recall. Using the same experimental settings as in the previous subsection, Figure 7 shows the precision and recall of our method in detecting true community dependencies among all relevant and retrieved pairs. The results clearly demonstrate that, in all cases, as the sample size increases, our method achieves near-perfect detection of ground-truth dependent community pairs.

## 6 Real Data Experiments

### 6.1 Enron Email Graphs

The Enron email dataset contains approximately $500,000$ emails generated by employees of the Enron Corporation (Priebe et al., 2010; Wang et al., 2014). We process the data into 18 binary undirected graphs, representing 5-week intervals from December 1999 to June 2002 [1]. There are a total of $n = 184$ email address, and we use a label vector with 11 classes based on the title of the employee, consisting of: 'Employee', 'Vice

---

[1] https://www.cis.jhu.edu/~parky/Enron/

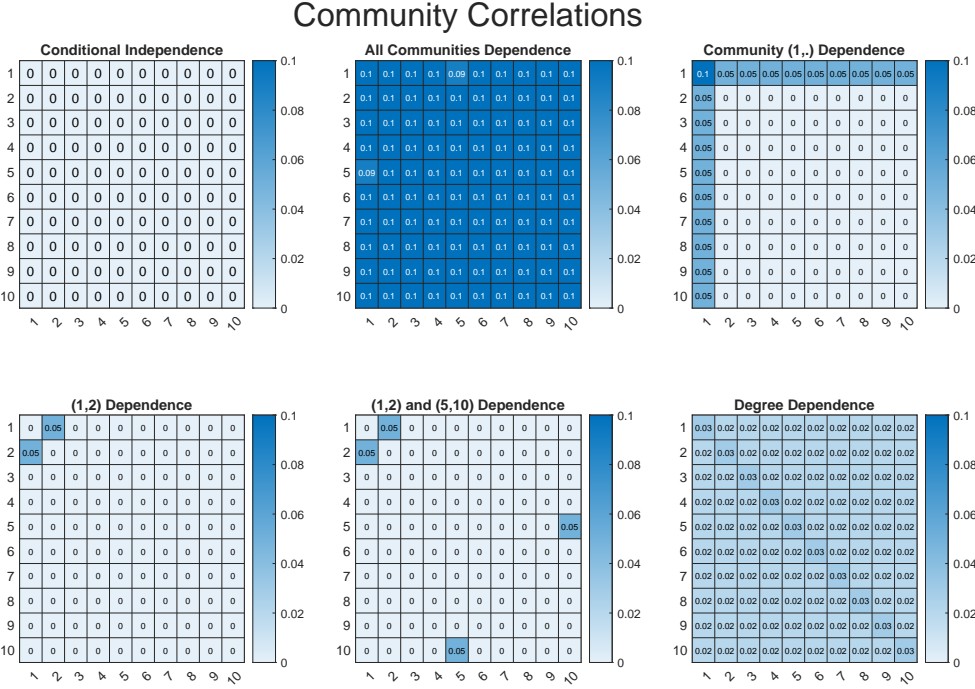

Figure 6: This figure visualizes the $10 \times 10$ community correlations throughout several graph dependence simulations.

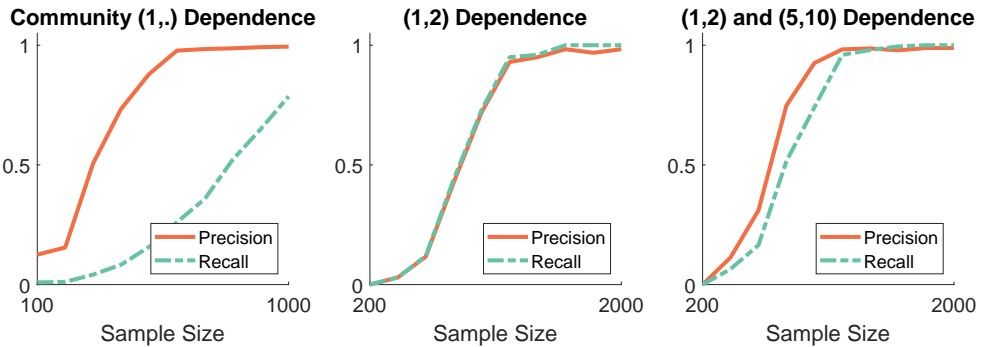

Figure 7: This figure plots the precision and recall for three graph dependence simulations.

President', 'President', 'Managing Director', 'Director', 'Manager', 'Trader', 'CEO', 'Lawyer', 'N/A', and 'xxx'.

Figure 8 shows the graph correlation and the maximum community correlation between each pair of graphs, all in absolute values. As the graph correlation tests unconditional independence, it shows a clear dependence structure as time progresses: graphs 1-10 are highly related to each other, but graphs 11-12 are relatively unrelated to others, and graphs 13-18 are highly related within themselves.

Tracing the timeline, graph 10 corresponds to August 2001, the period when the Enron CEO announced his resignation, while graph 13 corresponds to December 2001, right after Enron's bankruptcy announcement. Therefore, the graph correlation implies that the company's communication pattern was highly dependent

before the CEO resigned, drastically changed afterward, and then resumed a different dependence structure until it ceased operation. If we check the maximum community correlation, however, the correlation is much weaker — the same pattern still exists but to a much lesser degree. This behavior suggests that much of the graph correlation is due to the shared communities.

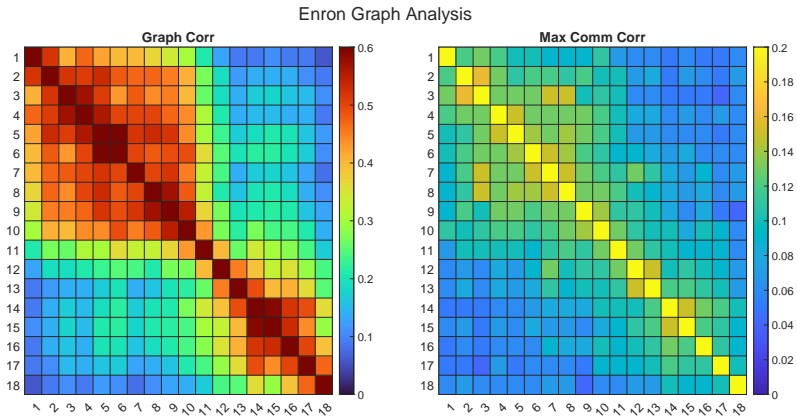

Figure 8: This figure shows the graph correlation and the maximum community correlation between the Enron email graphs.

## 6.2 Mouse Connectome Dataset

Here we apply graph correlation to mouse connectome networks. The data is open-access, processed into graphs (Chung et al., 2019), and available on GitHub [2], based on whole-brain diffusion magnetic resonance imaging-derived connectomes from four mouse lines (Wang et al., 2020). There are a total of 32 graphs from four mouse lines: BTBR, B6, CAST, DBA2, and for each line, there are eight age-matched mice. Each connectome was parcellated using a symmetric Waxholm Space, yielding a vertex set with a total of 332 regions of interest, i.e., there are $n = 332$ vertices for each of the 32 graphs. The 332 brain regions of interest can be divided into left and right hemispheres. Within each hemisphere, there are seven superstructures consisting of multiple regions, resulting in a total of $K = 14$ distinct communities. See (Gopalakrishnan et al., 2020) for a detailed analysis and visualization of this dataset.

We apply the graph correlation to the binarized graphs and report the graph correlation and the maximum community correlation between every pair of graphs in Figure 9. Similar to the Enron graphs, the graph correlation exhibits a strong correlation structure, while the maximum community correlation exhibits a much weaker correlation, implying much of the correlation comes from the shared communities. It also shows that mice within the same line have more related brain activity than mice from different lines: the BTBR mice are significantly different from all others, DBA2 and B6 are somewhat similar, and CAST are also different from others. This matches the underlying ground truth. For example, the BTBR mouse strain is a well-studied model that exhibits the core behavioral deficits that characterize autism spectrum disorders (ASD) in humans, which has significant neuroanatomical abnormalities, including the complete absence of the corpus callosum, a band of nerve fibers connecting the left and right hemispheres of the brain. Moreover, the B6, CAST, and DBA2 mice are all different mouse lines, with genetically distinct strains that do not exhibit ASD-like behaviors.

## 7 Conclusion

In this paper, we propose the notion of community correlations between two binary graphs, which is provably valid and consistent for testing unconditional dependence, as well as conditional independence when a label

---

[2]https://github.com/microsoft/graspologic

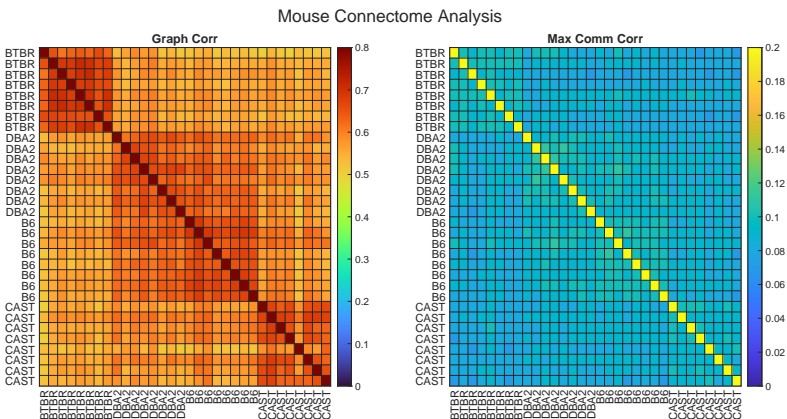

Figure 9: This figure shows the graph correlation and the maximum community correlation between the mouse connectome graphs.

vector is available. Our simulations and real data applications demonstrate both the practical utility and statistical advantages of these correlation measures.

While correlation and statistical dependence originate from classical statistics and information theory, they have been either core elements or have played crucial roles in a wide range of machine learning tasks, including kernel methods (Gretton et al., 2005b; Fukumizu et al., 2007), feature selection (Peng et al., 2005; Li et al., 2012), representation learning (Hjelm et al., 2019; Higgins et al., 2017), and causal discovery (Spirtes et al., 2000; Cai et al., 2022). For example, methods in representation learning often implicitly assume or enforce conditional independence between components of the learned representation, while algorithms in causal discovery use conditional independence testing to infer the underlying data-generating structure.

We believe the graph correlations introduced in this paper not only provide a solid theoretical foundation, but also enable new directions in graph-based machine learning involving multiple graphs. For example, in the setting of multiple matched graphs (Arroyo et al., 2021; Shen et al., 2024c), graph correlation can be used for graph selection, or to test conditional independencies for exploring causality. Moreover, this framework can potentially be extended to dynamic graph settings, where cross-correlations between graphs may be leveraged to further enhance graph embeddings and improve graph neural network learning (Yu et al., 2018; Shen et al., 2024b; Gallagher et al., 2021).

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

## APPENDIX

## A    Theorem Proofs

**Theorem 1.** *Given a pair of binary graph variables* $(A_1, A_2)$, *the following holds:*

- $\rho(k, l) = 0$ *if and only if* $A_1$ *and* $A_2$ *are conditionally independent on a specific pair* $(Y, Y') = (k, l)$.

- $\rho^m = 0$ *if and only if* $A_1$ *and* $A_2$ *are conditionally independent on all possible values of* $(Y, Y')$.

- $\rho = 0$ *if and only if* $A_1$ *and* $A_2$ *are unconditionally independent.*

*Proof.* (1) Recall that

$$\rho(k, l) = \frac{\Sigma_{12}(k, l)}{\sqrt{\Sigma_1(k, l)\Sigma_2(k, l)}}$$
$$\Sigma_{12}(k, l) = \text{Prob}(A_1 = 1, A_2 = 1|k, l) - \text{Prob}(A_1 = 1|k, l)\text{Prob}(A_2 = 1|k, l),$$

so $\rho(k, l) = 0$ if and only if $\Sigma_{12}(k, l) = 0$.

For the if direction: When $(A_1, A_2)$ are independent conditioned on $(Y, Y') = (k, l)$, it follows that

$$\text{Prob}(A_1, A_2|k, l) = \text{Prob}(A_1|k, l)\text{Prob}(A_2|k, l)$$

and thus $\Sigma_{12}(k, l) = 0$.

For the only if direction: $\Sigma_{12}(k, l) = 0$ means

$$\text{Prob}(A_1 = 1, A_2 = 1|k, l) = \text{Prob}(A_1 = 1|k, l)\text{Prob}(A_2 = 1|k, l).$$

It suffices to prove the equality holds for other values of $A_1$ and $A_2$, i.e., the equality holds when either or both of $A_1$ and $A_2$ are 0. We first check that:

$$\begin{aligned}
\text{Prob}(A_1 = 0, A_2 = 1|k, l) &= \text{Prob}(A_2 = 1|k, l) - \text{Prob}(A_1 = 1, A_2 = 1|k, l) \\
&= \text{Prob}(A_2 = 1|k, l) - \text{Prob}(A_1 = 1|k, l)\text{Prob}(A_2 = 1|k, l) \\
&= \text{Prob}(A_2 = 1|k, l)(1 - \text{Prob}(A_1 = 1|k, l)) \\
&= \text{Prob}(A_1 = 0|k, l)\text{Prob}(A_2 = 1|k, l).
\end{aligned}$$

Then, by using same argument but switching $A_1$ and $A_2$, we further obtain

$$\text{Prob}(A_1 = 1, A_2 = 0|k, l) = \text{Prob}(A_1 = 1|k, l)\text{Prob}(A_2 = 0|k, l).$$

Finally, it follows that

$$\begin{aligned}
\text{Prob}(A_1 = 0, A_2 = 0|k, l) &= \text{Prob}(A_2 = 0|k, l) - \text{Prob}(A_1 = 0, A_2 = 0|k, l) \\
&= \text{Prob}(A_2 = 0|k, l) - \text{Prob}(A_1 = 1|k, l)\text{Prob}(A_2 = 0|k, l) \\
&= \text{Prob}(A_2 = 0|k, l)(1 - \text{Prob}(A_1 = 1|k, l)) \\
&= \text{Prob}(A_1 = 0|k, l)\text{Prob}(A_2 = 0|k, l).
\end{aligned}$$

Therefore, when $\Sigma_{12}(k, l) = 0$, we always have

$$\text{Prob}(A_1, A_2|k, l) = \text{Prob}(A_1|k, l)\text{Prob}(A_2|k, l),$$

such that conditional independence holds.

(2) The maximum community correlation

$$\max_{k,l=1,\ldots,K}(|\rho(k,l)|) = 0$$

if and only if $\rho(k,l) = 0$ for all possible $(k,l)$. As $(Y,Y')$ is discrete with at most $K^2$ possible values, this immediately means $(A_1, A_2)$ are conditionally independent on all possible values of $(Y,Y')$.

(3) The derivation in part (1) holds regardless of whether the probabilities are conditioned on $(k,l)$ or not. By removing all conditioning on $(k,l)$, we immediately have $\rho = 0$ if and only if $A_1$ and $A_2$ are unconditionally independent.

$\square$

**Theorem 2.** *As the number of vertices $n \to \infty$, assume all $n_k$ also converge to infinity, and the threshold $\epsilon \to 0$. Then, all the sample covariance and variance estimators are consistent estimators of their population counterparts:*

$$\hat{\Sigma}_{12}(k,l) \stackrel{n\to\infty}{\to} \Sigma_{12}(k,l),$$
$$\hat{\Sigma}_1(k,l) \stackrel{n\to\infty}{\to} \Sigma_1(k,l),$$
$$\hat{\Sigma}_2(k,l) \stackrel{n\to\infty}{\to} \Sigma_2(k,l).$$

*As a result, the sample community correlation converges to the population correlation:*

$$\hat{\rho}(k,l) \stackrel{n\to\infty}{\to} \rho(k,l),$$

*the sample maximum correlation is a consistent estimator of the population maximum:*

$$\hat{\rho}^m \stackrel{n\to\infty}{\to} \rho^m,$$

*and the sample graph correlation also converges to the population graph correlation:*

$$\hat{\rho} \stackrel{n\to\infty}{\to} \rho.$$

*Proof.* Recall that

$$\hat{\Sigma}_{12}(k,l) = \sum_{\substack{i=1,\ldots,n}}^{\mathbf{Y}(i)=k} \frac{\mathbf{Z}_{12}(i,l)}{n_k} - \sum_{\substack{i=1,\ldots,n}}^{\mathbf{Y}(i)=k} \frac{\mathbf{Z}_1(i,l)}{n_k} \sum_{\substack{i=1,\ldots,n}}^{\mathbf{Y}(i)=k} \frac{\mathbf{Z}_2(i,l)}{n_k},$$
$$\Sigma_{12}(k,l) = \text{Prob}(A_1 = 1, A_2 = 1|k,l) - \text{Prob}(A_1 = 1|k,l)\text{Prob}(A_2 = 1|k,l).$$

Therefore, to prove that $\hat{\Sigma}_{12}(k,l) \to \Sigma_{12}(k,l)$, it suffices to prove

$$\sum_{\substack{i=1,\ldots,n}}^{\mathbf{Y}(i)=k} \frac{\mathbf{Z}_{12}(i,l)}{n_k} \to \text{Prob}(A_1 = 1, A_2 = 1|k,l)$$

$$\sum_{\substack{i=1,\ldots,n}}^{\mathbf{Y}(i)=k} \frac{\mathbf{Z}_1(i,l)}{n_k} \to \text{Prob}(A_1 = 1|k,l),$$

$$\sum_{\substack{i=1,\ldots,n}}^{\mathbf{Y}(i)=k} \frac{\mathbf{Z}_2(i,l)}{n_k} \to \text{Prob}(A_1 = 2|k,l).$$

The graph encoder embedding satisfies $\mathbf{Z}_1(i,l) = \sum_{j=1,\ldots,n}^{\mathbf{Y}(j)=l} \mathbf{A}_1(i,j)/n_l$, so it follows that

$$\sum_{i=1,\ldots,n}^{\mathbf{Y}(i)=k} \frac{\mathbf{Z}_{12}(i,l)}{n_k} = \sum_{i=1,\ldots,n}^{\mathbf{Y}(i)=k} \sum_{j=1,\ldots,n}^{\mathbf{Y}(j)=l} \frac{\mathbf{A}_1(i,j)\mathbf{A}_2(i,j)}{n_k n_l}$$

$$\sum_{i=1,\ldots,n}^{\mathbf{Y}(i)=k} \frac{\mathbf{Z}_1(i,l)}{n_k} = \sum_{i=1,\ldots,n}^{\mathbf{Y}(i)=k} \sum_{j=1,\ldots,n}^{\mathbf{Y}(j)=l} \frac{\mathbf{A}_1(i,j)}{n_k n_l},$$

$$\sum_{i=1,\ldots,n}^{\mathbf{Y}(i)=k} \frac{\mathbf{Z}_2(i,l)}{n_k} = \sum_{i=1,\ldots,n}^{\mathbf{Y}(i)=k} \sum_{j=1,\ldots,n}^{\mathbf{Y}(j)=l} \frac{\mathbf{A}_2(i,j)}{n_k n_l}.$$

As we assumed $(\mathbf{A}_1(i,j), \mathbf{A}_2(i,j)) \overset{i.i.d.}{\sim} (A_1, A_2)$ for $i < j$, and $(A_1, A_2)$ are binary for each, by law of large numbers,

$$\sum_{i=1,\ldots,n}^{\mathbf{Y}(i)=k} \frac{\mathbf{Z}_{12}(i,l)}{n_k} \to E(A_1 A_2 | Y = k, Y' = l) = Prob(A_1 = 1, A_2 = 1 | k, l),$$

$$\sum_{i=1,\ldots,n}^{\mathbf{Y}(i)=k} \frac{\mathbf{Z}_1(i,l)}{n_k} \to E(A_1 | Y = k, Y' = l) = Prob(A_1 = 1 | k, l),$$

$$\sum_{i=1,\ldots,n}^{\mathbf{Y}(i)=k} \frac{\mathbf{Z}_2(i,l)}{n_k} \to E(A_2 | Y = k, Y' = l) = Prob(A_2 = 1 | k, l).$$

Applying the above convergence to all covariance and variance terms, we immediately have

$$\hat{\Sigma}_{12}(k,l) \overset{n\to\infty}{\Rightarrow} \Sigma_{12}(k,l),$$
$$\hat{\Sigma}_1(k,l) \overset{n\to\infty}{\Rightarrow} \Sigma_1(k,l),$$
$$\hat{\Sigma}_2(k,l) \overset{n\to\infty}{\Rightarrow} \Sigma_2(k,l).$$

Excluding the trivial case that the denominator equals 0, the above convergence leads to the convergence of sample community correlation. When all sample community correlations converge to the population correlations, the sample maximum correlation also converges to the population maximum. Finally, the sample graph correlation converges as well, which is simply a special case of community correlation where $k = l = K = 1$. $\qquad\square$

**Theorem 3.** *Let $n$ increase to infinity, and assume the underlying binary graph variables $(A_1, A_2)$ are conditionally independent on $(Y, Y')$. When $k \neq l$, the sample community correlation satisfies*

$$\sqrt{n_k n_l} \hat{\rho}(k,l) \overset{dist}{\Rightarrow} Normal(0,1)$$

*with a difference of at most $O(\frac{1}{n})$, i.e., denote $U = \sqrt{n_k n_l} \hat{\rho}(k,l)$ and $V$ be the standard normal variable, their distribution satisfies:*

$$|f_U(x) - f_V(x)| = O(\frac{1}{n}).$$

*When $k = l$, the sample community correlation satisfies the same, except it converges to $Normal(0, \sqrt{2})$, i.e.,*

$$n_k \hat{\rho}(k,k) \overset{dist}{\Rightarrow} Normal(0, \sqrt{2}).$$

*Proof.* Recall that

$$\hat{\Sigma}_{12}(k,l) = \sum_{i=1,\dots,n}^{\mathbf{Y}(i)=k} \frac{\mathbf{Z}_{12}(i,l)}{n_k} - \sum_{i=1,\dots,n}^{\mathbf{Y}(i)=k} \frac{\mathbf{Z}_1(i,l)}{n_k} \sum_{i=1,\dots,n}^{\mathbf{Y}(i)=k} \frac{\mathbf{Z}_2(i,l)}{n_k},$$

$$\hat{\Sigma}_1(k,l) = \sum_{i=1,\dots,n}^{\mathbf{Y}(i)=k} \frac{\mathbf{Z}_1(i,l)}{n_k} - \sum_{i=1,\dots,n}^{\mathbf{Y}(i)=k} \frac{\mathbf{Z}_1(i,l)}{n_k} \sum_{i=1,\dots,n}^{\mathbf{Y}(i)=k} \frac{\mathbf{Z}_1(i,l)}{n_k},$$

$$\hat{\Sigma}_2(k,l) = \sum_{i=1,\dots,n}^{\mathbf{Y}(i)=k} \frac{\mathbf{Z}_2(i,l)}{n_k} - \sum_{i=1,\dots,n}^{\mathbf{Y}(i)=k} \frac{\mathbf{Z}_2(i,l)}{n_k} \sum_{i=1,\dots,n}^{\mathbf{Y}(i)=k} \frac{\mathbf{Z}_2(i,l)}{n_k},$$

and

$$\sqrt{n_k n_l}\hat{\rho}(k,l) = \frac{\sqrt{n_k n_l}\hat{\Sigma}_{12}(k,l)}{\sqrt{\hat{\Sigma}_1(k,l)\hat{\Sigma}_2(k,l)}}.$$

From the proof of Theorem 2, the denominator converges to

$$\sqrt{\hat{\Sigma}_1(k,l)\hat{\Sigma}_2(k,l)} \to \sqrt{\Sigma_1(k,l)\Sigma_2(k,l)}$$

at the rate of $O(\frac{1}{n})$, and the numerator satisfies

$$\sqrt{n_k n_l}\hat{\Sigma}_{12}(k,l) = \sum_{i=1,\dots,n}^{\mathbf{Y}(i)=k}\sum_{j=1,\dots,n}^{\mathbf{Y}(j)=l} \frac{\mathbf{A}_1(i,j)\mathbf{A}_2(i,j)}{\sqrt{n_k n_l}} - \sum_{i=1,\dots,n}^{\mathbf{Y}(i)=k}\sum_{j=1,\dots,n}^{\mathbf{Y}(j)=l} \frac{\mathbf{A}_1(i,j)}{\sqrt{n_k n_l}} \sum_{i=1,\dots,n}^{\mathbf{Y}(i)=k}\sum_{j=1,\dots,n}^{\mathbf{Y}(j)=l} \frac{\mathbf{A}_2(i,j)}{n_k n_l}.$$

Denote $p = Prob(A_2 = 1|k,l)$ and

$$\gamma = \sum_{i=1,\dots,n}^{\mathbf{Y}(i)=k}\sum_{j=1,\dots,n}^{\mathbf{Y}(j)=l} \frac{\mathbf{A}_1(i,j)(\mathbf{A}_2(i,j) - p)}{\sqrt{n_k n_l}},$$

we observe that

$$|\sqrt{n_k n_l}\hat{\Sigma}_{12}(k,l) - \gamma| = \sum_{i=1,\dots,n}^{\mathbf{Y}(i)=k}\sum_{j=1,\dots,n}^{\mathbf{Y}(j)=l} \frac{\mathbf{A}_1(i,j)}{\sqrt{n_k n_l}} \Big( \sum_{i=1,\dots,n}^{\mathbf{Y}(i)=k}\sum_{j=1,\dots,n}^{\mathbf{Y}(j)=l} \frac{\mathbf{A}_2(i,j)}{n_k n_l} - p \Big)$$

$$= O(\frac{1}{n^2}) \to 0.$$

This is because the the term within the bracket converges to 0 at the rate of $O(\frac{1}{n^2})$ by the law of large numbers as in the proof of Theorem 2, and the term outside the bracket is normally distributed by the central limit theorem with mean being $Prob(A_1 = 1|k,l)$ and variance bounded by $1/4$.

Applying central limit theorem to $\gamma$: as $n \to \infty$, $\gamma$ is normally distributed, for which the mean equals 0, and the variance equals

$$Var(\gamma) = Var(A_1|k,l)Var(A_2|k,l)$$
$$= \mathrm{Prob}(A_1 = 1|k,l)(1 - \mathrm{Prob}(A_1 = 1|k,l))\mathrm{Prob}(A_2 = 1|k,l)(1 - \mathrm{Prob}(A_2 = 1|k,l))$$
$$= \Sigma_1(k,l)\Sigma_2(k,l).$$

Finally, as $n \to \infty$,

$$\rho(k,l) \to \frac{\gamma}{\sqrt{\Sigma_1(k,l)\Sigma_2(k,l)}} \sim \mathrm{Normal}(0,1),$$

as the variance of $\gamma$ cancels out by the denominator.

Now, the above proof works for the case of $k \neq l$, or when $k = l$ but the graph is directed. In case of $k = l$ and undirected graphs, we shall note that the numerator has repeated entries due to the symmetry of the graph adjacency submatrix, i.e.,

$$\gamma = \sum_{i=1,\ldots,n}^{\mathbf{Y}(i)=k} \sum_{j=1,\ldots,n}^{\mathbf{Y}(j)=k} \frac{\mathbf{A}_1(i,j)(\mathbf{A}_2(i,j) - p)}{n_k}$$

$$= \sum_{i=1,\ldots,n}^{\mathbf{Y}(i)=k} \sum_{j>i}^{\mathbf{Y}(j)=k} \frac{2\mathbf{A}_1(i,j)(\mathbf{A}_2(i,j) - p)}{n_k}.$$

As there are a total of $n_k^2/2$ terms in the numerator, computing the variance of $\gamma$ ends up being $2\Sigma_1(k,l)\Sigma_2(k,l)$. After normalization by the denominator, $\rho(k,k)$ is asymptotically normally distributed with mean 0 and variance 2.

A small detail we omitted above is the diagonal entries, which are always 0 when $k = l$. However, there are only $n_k$ diagonal entries comparing to $n_k^2 - n_k$ other adjacency entries, therefore the diagonal entries can be omitted for asymptotic purposes and have a negligible difference of at most $O(\frac{1}{n})$. $\qquad\square$

