# OpenReview forum: "Community Correlations and Testing Independence Between Binary Graphs"
_TMLR — Rejected by TMLR_

### Review · Reviewer_rKDh · 2025-03-17

**Summary Of Contributions:**

The paper reviews the correlation and dependence between the distribution of two binary graphs.  The paper explores vertex of communities and their correlation coefficients. Simulations were performed and reported as well.

While the paper presents some contributions on the simulation of graphs and the extraction of their correlation, I don't see it as a good fit for TMLR.  In its current form, this paper should be published in another venue more suitable for graph-distribution and simulations than a ML venue.

**Audience:**

No

**Broader Impact Concerns:**

No ethical considerations were presented in the paper.

**Claims And Evidence:**

No

**Requested Changes:**

- The paper should be re-written to better explain the proposal beyond the theoretical considerations.  In specific, how the proposal fits the ML graph problems and how these coefficients can be used in ML tasks.
- The algorithms should be supported with pseudo-code to help understand the main contributions.
- The relation of the proposal and ML should be further explored.
- There should be a discussion and conclusion section to close the paper.
- The  $I$ function is not defined when introduced.
- Typo in P4 "The motivates"

As mentioned in the contributions, the paper seems to be unsuitable for TMLR.  The changes that this paper requires to be of interest to the machine learning audience are major and are beyond a discussion that can be performed during the review process.  **I strongly suggest the authors to find a more suitable venue for their work** rather than hastily trying to perform additional experiments to evaluate the proposal and make it fit into a ML venue.

**Strengths And Weaknesses:**

**Strengths:**
- The paper explores an interesting problem of identifying the correlation between two communities
- The paper performs some simulations.

**Weaknesses:**
- It is not clear how the proposal fits the Machine Learning community.  While graphs and their embeddings are extensively used in ML, the paper is limited to the exploration of the correlation between the graphs and the creation of distributions where these graphs could be simulated from.
- The link between the distributions and their use in the ML literature deteriorates rather fast beyond the introduction.
- The descriptions of the graph distributions and their usage could be improved significantly.  It is hard to follow how these ideas are used within the simulations.
- The paper ends abruptly and has no discussion on the impact of the results nor conclusions.

---

> ### Author Response · Authors · 2025-04-13
>
> Thank you for reviewing our paper and providing constructive feedback. We believe we have addressed most of the points raised in your comments, including:
>
> 1. Minor issues have been corrected, including the phrase on page 4 (“This motivates...”), and the definition of the indicator function $I(\cdot)$, which is now introduced before its first use in the equation. Additionally, the working code—including correlation computation, simulations, and real data analysis—will be made publicly available on GitHub. The link is currently anonymized in the manuscript to preserve anonymity during review.
>
> 2. We have added a conclusion section (pages 16–17) specifically to address your concern about the relevance of correlation and dependence in machine learning, and to clarify how our proposed graph correlation measure can contribute to graph learning problems.
>
> Technically, concepts such as correlation and statistical dependence originate from classical statistics and information theory. While they are not machine learning algorithms in themselves, they serve as foundational tools that are central to many areas of modern machine learning. In the conclusion, we provide several examples—supported by highly cited papers from leading machine learning journals and conferences—where these concepts are either core components or play a critical role in the design of machine learning algorithms, including:
>
> -- Kernel Methods and Kernel Correlation: These techniques measure similarity between data distributions using kernel-based dependence measures. The Hilbert-Schmidt Independence Criterion (HSIC) is a popular kernel-based statistic for assessing dependence between random variables. It is widely used in independence testing, two-sample testing, and causal discovery [1,2].
>
> -- Feature Selection: The goal is to select features most relevant to the target while minimizing redundancy, a problem framed as maximizing dependence with the label and minimizing dependence among features [3,4].
>
> -- Representation Learning: Many methods aim to learn embeddings where certain variables (e.g., latent codes) are disentangled. This often implicitly assumes or enforces conditional independence between parts of the representation, given the input [5,6].
>
> -- Causal Discovery: Causal structure can be inferred from observational data using conditional independence as a core criterion. Many algorithms systematically test for conditional independencies among observed variables to recover causal graphs [7,8].
>
> In the newly added conclusion section, we summarized the value of correlation and dependence in those areas, supported by the citations above. We also discuss the applicability of our proposed graph correlation measure to tasks involving multiple graphs, with potential applications in multi-graph embedding [9,10] and dynamic graph neural networks [11].
>
> We hope the addition of the conclusion section, along with the detailed explanation above, helps to address your concern regarding the relevance and value of correlation and dependence in the context of machine learning.
>
>
> References:
>
> 1. Gretton et al., Measuring statistical dependence with Hilbert-Schmidt norms, ALT 2005
>
> 2. Fukumizu et al., Kernel Measures of Conditional Dependence, NIPS 2007
>
> 3. Peng et al., Feature selection based on mutual information criteria of max-dependency, max-relevance, and min-redundancy, IEEE TPAMI 2005
>
> 4. Li et al., Feature Screening via Distance Correlation Learning, JASA 2012.
>
> 5. Hjelm et al., Learning Deep Representations by Mutual Information Estimation and Maximization, ICLR 2019
>
> 6. Higgins et al., beta-VAE: Learning Basic Visual Concepts with a Constrained Variational Framework, ICLR 2017
>
> 7. Spirtes et al., Causation, Prediction, and Search, MIT Press, 2000
>
> 8. Cai et al., A Distribution Free Conditional Independence Test with Applications to Causal Discovery, JMLR 2022
>
> 9. Arroyo et al., Inference for multiple heterogeneous networks with a common invariant subspace. JMLR 2021
>
> 10. Shen et al., Synergistic graph fusion via encoder embedding. Information Sciences 2024
>
> 11. Yu et al., Spatio-Temporal Graph Convolutional Networks: A Deep Learning Framework for Traffic Forecasting, IJCAI 2018

---

> > ### Comment · Reviewer_rKDh · 2025-04-24
> >
> > I thank the authors for the update to the conclusions and the reply.  However, as stated in my original review, I think that the problems are more foundational than the minor edits made to the paper.

---

### Review · Reviewer_vkAR · 2025-03-22

**Summary Of Contributions:**

This paper proposes a novel set of correlation measures—termed community correlation, maximum community correlation, and graph correlation—for quantifying dependence between two binary graphs. Building on the stochastic block model, the authors define community correlation as a measure that is zero if and only if the two graphs are conditionally independent on a given pair of vertex communities. Extending this idea, the maximum community correlation is introduced to test conditional independence across all community pairs, while the graph correlation assesses unconditional independence. The paper develops a sample version of these statistics using a graph encoder embedding method and rigorously proves convergence properties and the asymptotic null distributions. Extensive simulations under both the standard SBM and its degree-corrected variant (DC-SBM) illustrate the validity and power of the proposed tests. Finally, the method is demonstrated on two real datasets—the Enron email network and mouse connectome graphs—highlighting its practical utility in capturing both shared community structure and genuine dependence beyond that structure.

**Audience:**

Yes

**Broader Impact Concerns:**

No specific concerns

**Claims And Evidence:**

Yes

**Requested Changes:**

* A conclusion should be included to summarize the key takeaways of the paper after all the discussions.
*  A discussion section is suggested to be included to identify the border impacts to other related topics (e.g., graph neural networks or deep learning, or any border topics the authors find worth discussion).
* Codes are strongly encouraged to be included to check the correctness of implementations.
* It would also be interesting to compare the FDR in the simulation study
* A diagram would be helpful to understand the method, the current version is a bit dry.

**Strengths And Weaknesses:**

**Strengths**
* The paper establishes clear theoretical guarantees (e.g., consistency of the estimators and asymptotic null distributions) via well-formulated theorems, providing strong statistical backing for the proposed methods.
* This work is well-motivated and novel. It well-addresses the drawbacks of the conventional spectural embedding approach in graph ML.By introducing the idea of community correlation, the work directly extends classical correlation concepts to complex graph data, addressing the unique challenges of non-i.i.d. data inherent in graphs.
* The authors conduct extensive simulations under various stochastic block model settings, including both standard and degree-corrected variants, which convincingly demonstrate the validity and testing power of the methods.
* Complexity is well-established

**Weaknesses**
* Conclusion is missing
* Codes are encouraged to be available for reproducibility
* A discussion section is suggested to identify potential extensions and future works.

---

> ### Author Response · Authors · 2025-04-13
>
> Thank you for reviewing our paper! We have revised the manuscript to address your comments as follows:
>
> 1. We have added a conclusion section (pages 16–17) to summarize the key contributions of the paper. The section highlights the usefulness of general correlation measures across multiple machine learning tasks, including feature selection [1,2], representation learning [3,4], and causal discovery [5,6]. We also discuss the broader applicability of our proposed graph correlation measure to any graph learning task involving multiple graphs, with potential applications in multi-graph embedding [7,8] and dynamic graph neural networks [9].
>
> 2. We have compiled the code and will make it publicly available on GitHub. The repository includes implementations for correlation computation, simulations, and real data analysis. For the purpose of anonymous review, the GitHub link is currently anonymized in the manuscript.
>
> 3. On pages 14–15, we have added a new subsection and Figure 7 to evaluate the precision and recall of our method in correctly identifying ground-truth pairs with significant community correlation. Excluding trivial settings in which all pairs are dependent or independent, Figure 7 demonstrates that our method achieves near-perfect precision and recall as the sample size increases.
>
> 4. The proposed sample correlation method estimates the relationship between $A_1$ and $A_2$, which is conceptually similar to Pearson correlation but involves double summations over vertex pairs due to the graph adjacency structure. For this reason, we found it difficult to design a visual diagram for the computation.
>
> Instead, to enhance the paper’s clarity and appeal, we have added Section 2.3 and Figure 1 on pages 4–5. This section uses standard random variables and Pearson correlation to motivate the need for both unconditional and conditional correlations—which correspond in our setting to overall graph correlation and community correlation, respectively.
>
> References:
>
> 1. Peng et al., Feature selection based on mutual information criteria of max-dependency, max-relevance, and min-redundancy, IEEE TPAMI 2005
>
> 2. Li et al., Feature Screening via Distance Correlation Learning, JASA 2012.
>
> 3. Hjelm et al., Learning Deep Representations by Mutual Information Estimation and Maximization, ICLR 2019
>
> 4. Higgins et al., beta-VAE: Learning Basic Visual Concepts with a Constrained Variational Framework, ICLR 2017
>
> 5. Spirtes et al., Causation, Prediction, and Search, MIT Press, 2000
>
> 6. Cai et al., A Distribution Free Conditional Independence Test with Applications to Causal Discovery, JMLR 2022
>
> 7. Arroyo et al., Inference for multiple heterogeneous networks with a common invariant subspace. JMLR 2021
>
> 8. Shen et al., Synergistic graph fusion via encoder embedding. Information Sciences 2024
>
> 9. Yu et al., Spatio-Temporal Graph Convolutional Networks: A Deep Learning Framework for Traffic Forecasting, IJCAI 2018

---

### Review · Reviewer_L33K · 2025-03-31

**Summary Of Contributions:**

The paper gives formulas for correlation between graphs based on stochastic block models.  It then gives estimators for the formulated correlations and proves their consistency and asymptotic normality.  This allows for the construction of confidence intervals.  The estimators are then applied to real graph datasets.

**Audience:**

Yes

**Claims And Evidence:**

Yes

**Requested Changes:**

1.) On page 3, in the last display equation in Section 2.1, the indexing of the summation seems incorrect --  the variables should only take values up to $K$.  The same holds for the remaining display equations on the same page.

2.) At the top of page 4, I do not understand how two graphs can be unconditionally independent if their community indicators are dependent.  This statement also appears to contradict the last statement on the previous page.  Can you please clarify this?

3.) Definition 1 is a little bit imprecise: it appears to define conditional independence between a pair of arbitrary graphs, but I believe the intent is for it to apply to pairs of graph-valued random variables.  The same applies to Definition 2.  Please clarify.

4.) I feel that somehow the text on page 4 could be condensed by acknowledging that Definition 2 is simply the standard notion of probabilistic independence.


5.) Would it be of interest to consider a case where the number of communities tends to $\infty$ as $n\to\infty$?

**Strengths And Weaknesses:**

Strengths:

1.) The theoretical analysis for estimators of graph correlation is fairly comprehensive, including convergence in distribution results for the proposed estimators.

2.) The estimators are applied to both synthetic and real datasets.

3.) The work fits into the general topic of statistics on graphs, which is of interest to a significant subset of the TMLR audience.


Weaknesses:

1.) There are several typos and notational oddities (see below, in the "Requested changes" section).

2.) The definitions of correlation (conditional and unconditional) are simple adaptations of standard probability notions.

3.) The paper does not really convey any sort of technical challenge in the analysis.

4.) The asymptotic analysis only seems to address the case where the number of communities is $O(1)$ as $n\to\infty$.

---

> ### Author Response · Authors · 2025-04-13
>
> Thank you for reviewing our paper. We have made the following changes in response to your comments:
>
> 1. On page 3, the summation index has been corrected to $K$ in all relevant equations.
>
> 2. Thank you for pointing this out. This was indeed a typo. We have revised the sentence to: "...when two graphs have the same or dependent vertex communities, unconditional independence becomes trivially false."
>
> 3 & 4. You are absolutely right—the notation is intended to represent pairs of graph-valued random variables. Specifically, $A_1$ denotes the edge value between two vertices in one graph, and $A_2$ denotes the edge value between the same two vertices in another graph. These variables are not fundamentally different from standard random variables, except that they are indexed by two underlying vertices drawn from communities $Y$ and $Y'$, respectively.
>
> To clarify this, we have added a sentence after Definition 2 on page 4: "Note that both definitions are grounded in the standard notion of probabilistic independence, but are formulated in terms of a pair of graph-valued random variables."
>
> 5. The proposed method itself does not change with the value of $K$, and in the worst-case scenario, $K = n$. However, the value of $K$ can affect the computational complexity. We have updated Section 4.1.7 on page 9 to explicitly discuss this case and the computational implications.
>
> "...Since sparse graphs are common in social networks, the average complexity is $O(nK + s + K^2)$. Given that $K$ is typically a small constant, the number of edges $s$ becomes the dominant term, simplifying the average complexity to $O(s)$ for sparse graphs. For dense graphs or when $K$ is on the order of $O(n)$, the worst-case complexity reaches $O(n^2)$..."

---

### Review · Reviewer_9KhJ · 2025-03-31

**Summary Of Contributions:**

This paper introduces community correlations to measure edge associations within vertex communities. The proposed framework proposes maximum community correlation for testing conditional independence across all community pairs and overall graph correlation for assessing unconditional independence. The method is validated through simulations and two real datasets.

**Audience:**

Yes

**Claims And Evidence:**

No

**Requested Changes:**

The authors can find the suggested changes highlighted in the weaknesses section.

**Strengths And Weaknesses:**

Strength 1: This paper proposes a method for testing the community correlation of two graphs based on the correlation coefficient, along with theoretical justification. For conditional independence testing, the proposed method is intuitive and reasonable.

Weakness 1: The paper lacks a clear motivation for testing community correlation. A concrete motivating example is needed. While TMLR primarily focuses on correctness, incorporating a real machine learning problem to illustrate the relevance of the proposed method would strengthen the paper’s justification.

Weakness 2: The authors discuss both unconditional and conditional independence, but their presentation at the beginning of the paper is somewhat confusing. A graphical illustration highlighting the difference between these two concepts in the introduction would greatly enhance clarity.

Weakness 3: The graph correction for unconditional independence is unclear, particularly in estimating the probability $P(A_1 = 1, A_2 = 1)$. For instance, given two $n \times n $ networks, permuting the nodes of any network can lead to different empirical estimates of $P(A_1 = 1, A_2 = 1)$. In other words, I feel the third point of Theorem 1 might not be true or requires more clarifications and justifications.

---

> ### Author Response · Authors · 2025-04-13
>
> Thank you for reviewing our paper! We have revised the paper to address your points. Specifically, for the requested three weakness:
>
> 1) We have added a conclusion section (page 16-17) to better highlight and clarify the connection between the graph correlation proposed in this paper and its potential utility.
>
> Specifically, we acknowledge that correlation and dependence originate from classical statistics and information theory, but they serve as foundational tools in many machine learning tasks. To support this point, we now cite several highly cited papers from top machine learning journals and conferences that are either directly focused on correlation and dependence—such as kernel correlation and conditional kernel correlation [1,2]—or use correlation and conditional dependence as a key component, including in feature selection [3,4], representation learning [5,6], and causal discovery [7,8].
>
> In the conclusion section, we further emphasize the potential of our method to contribute to any graph learning task involving multiple graphs. For instance, to enhance multi-graph embeddings [9,10], or to support cross-correlation analysis in dynamic graph neural networks [11].
>
> 2. On pages 4–5, we added Section 2.3 and Figure 1 to illustrate the necessity of computing both unconditional and conditional correlations, i.e., overall graph correlation and community correlation. From a theoretical perspective, computing both gives a more complete picture of the correlation structure: conditional correlation can be masked after averaging across subgroups, while unconditional correlation can appear significant simply due to shared community structure. From a practical standpoint, it is useful to define and compute both types of correlation, as many machine learning algorithms rely on conditional dependence, such as those used in causal discovery.
>
> 3. Yes, permuting the node labels can lead to different empirical estimates. However, our setting explicitly assumes two matched graphs, and no permutation is involved in either the correlation computation or the testing procedure. If one were to randomly permute the vertices in one graph, the resulting pair of graphs would naturally become independent (both unconditionally and conditionally), as the correspondence between nodes would be broken. Therefore we believe this point shall be irrelevant --- if we misunderstood your point, please kindly let us know.
>
>
> References:
>
> 1. Gretton et al., Measuring statistical dependence with Hilbert-Schmidt norms, ALT 2005
>
> 2. Fukumizu et al., Kernel Measures of Conditional Dependence, NIPS 2007
>
> 3. Peng et al., Feature selection based on mutual information criteria of max-dependency, max-relevance, and min-redundancy, IEEE TPAMI 2005
>
> 4. Li et al., Feature Screening via Distance Correlation Learning, JASA 2012.
>
> 5. Hjelm et al., Learning Deep Representations by Mutual Information Estimation and Maximization, ICLR 2019
>
> 6. Higgins et al., beta-VAE: Learning Basic Visual Concepts with a Constrained Variational Framework, ICLR 2017
>
> 7. Spirtes et al., Causation, Prediction, and Search, MIT Press, 2000
>
> 8. Cai et al., A Distribution Free Conditional Independence Test with Applications to Causal Discovery, JMLR 2022
>
> 9. Arroyo et al., Inference for multiple heterogeneous networks with a common invariant subspace. JMLR 2021
>
> 10. Shen et al., Synergistic graph fusion via encoder embedding. Information Sciences 2024
>
> 11. Yu et al., Spatio-Temporal Graph Convolutional Networks: A Deep Learning Framework for Traffic Forecasting, IJCAI 2018

---

### Decision · Action_Editor_WgUs · 2025-05-14

**Recommendation:** Reject

**Comment:**

This work provides good theory for measuring community correlation between two graphs, particularly in the conditionally independent case. It is suggested that the case for the unconditional independence test should be strengthened. More experiments, for example on ultra-large graph data, would greatly improve the work as well.

**Audience:**

Yes.

**Claims And Evidence:**

Assuming two graphs are generated from the same stochastic block model, the authors propose several quantities to test conditional and unconditional independence between the two graphs.

+) The proposed correlations are shown to be zero iff (un)conditional independence holds and is consistent as the number of vertices goes to infinity.

-) The practical meaning of, specifically the difference between, conditional and unconditional independence is unclear. In fact, unconditional independence is not even validated on real data, which is an issue that should be addressed more carefully in the revision.

**Resubmission Of Major Revision:**

The authors may consider submitting a major revision at a later time.